# Mapping Plastic Mulched Farmland for High Resolution Images of Unmanned Aerial Vehicle Using Deep Semantic Segmentation

**Qinchen Yang [1,2,3], Man Liu [1,2,3], Zhitao Zhang [4], Shuqin Yang [2,3,5], Jifeng Ning [1,2,3,*]** and **Wenting Han [2,3,6]**

1   College of Information Engineering, Northwest A&F University, Yangling 712100, China
2   Key Laboratory of Agricultural Internet of Things, Ministry of Agriculture and Rural Affairs,
    Yangling 712100, China
3   Shaanxi Key Laboratory of Agricultural Information Perception and Intelligent Service,
    Yangling 712100, China
4   College of Water Resources and Architectural Engineering, Northwest A&F University,
    Yangling 712100, China
5   College of Mechanical and Electronic Engineering, Northwest A&F University, Yangling 712100, China
6   Institute of Water Saving Agriculture in Arid Regions of China, Yangling 712100, China
*   Correspondence: njf@nwsuaf.edu.cn; Tel.: +86-150-9185-7187

**Abstract:** With increasing consumption, plastic mulch benefits agriculture by promoting crop quality and yield, but the environmental and soil pollution is becoming increasingly serious. Therefore, research on the monitoring of plastic mulched farmland (PMF) has received increasing attention. Plastic mulched farmland in unmanned aerial vehicle (UAV) remote images due to the high resolution, shows a prominent spatial pattern, which brings difficulties to the task of monitoring PMF. In this paper, through a comparison between two deep semantic segmentation methods, SegNet and fully convolutional networks (FCN), and a traditional classification method, Support Vector Machine (SVM), we propose an end-to-end deep-learning method aimed at accurately recognizing PMF for UAV remote sensing images from Hetao Irrigation District, Inner Mongolia, China. After experiments with single-band, three-band and six-band image data, we found that deep semantic segmentation models built via single-band data which only use the texture pattern of PMF can identify it well; for example, SegNet reaching the highest accuracy of 88.68% in a 900 nm band. Furthermore, with three visual bands and six-band data (3 visible bands and 3 near-infrared bands), deep semantic segmentation models combining the texture and spectral features further improve the accuracy of PMF identification, whereas six-band data obtains an optimal performance for FCN and SegNet. In addition, deep semantic segmentation methods, FCN and SegNet, due to their strong feature extraction capability and direct pixel classification, clearly outperform the traditional SVM method in precision and speed. Among three classification methods, SegNet model built on three-band and six-band data obtains the optimal average accuracy of 89.62% and 90.6%, respectively. Therefore, the proposed deep semantic segmentation model, when tested against the traditional classification method, provides a promising path for mapping PMF in UAV remote sensing images.

**Keywords:** plastic mulched farmland; fully convolutional networks; unmanned aerial vehicle remote sensing image; deep semantic segmentation

---

## 1. Introduction

Plastic mulching is a method of covering the farmland surface with agricultural plastic membrane of different thicknesses and colors and at different intervals. It can stimulate the growth of seedlings and help low-yield or no-yield wasteland to be used for modern agricultural activities by reducing soil moisture evaporation, preventing diseases and pests and maintaining temperature [1]. It also has the advantages of low cost and convenient use. Hence, plastic mulching technology is extensively applied, and China is the largest consumer of plastic film geographically, consuming more than 1 million tons per year [2].

However, the widespread use of plastic mulch results in a lot of environmental problems. The plastic film is difficult to decompose rapidly. A huge amount of residues have caused a reduction in land renewable ability and pollution to water and soil, which in turn affect the production of crops [3]. To solve these problems, the relevant government departments and enterprises are urgently looking for solutions with experts pushing measures to raise the amount of plastic film recovery [4]. These measures require accurate spatial and temporal data on plastic film distribution. Traditional methods for plastic mulched farmland (PMF) recognition by remote sensing images require manual extraction of features. Yet, different spectral and texture features have a great impact on the classification results. In 2016, Hasituya et al. [5] compared the accuracies of spectral and texture features from Landsat-8 Operational Land Imager (OLI) images and found that spectral features contribute more to plastic film, and its texture features produce only a few improvements. In their subsequent work [6], backscattering intensity of different polarizations and multiple polarimetric decomposition descriptors were proved to be suitable information for PMF mapping. Lu et al [7] mapped PMF by decision tree classifier for Landsat-5 satellite images in 1998, 2007, and 2011, which has the characteristic of being temporally stable and is suitable for PMF recognition in different time periods of a large area. After which, they used a threshold model and moderate-resolution imaging spectroradiometer time series data to detect transparent PMF for continental and global scales [8]. There are also some other works on PMF recognition which were done, such as selecting optimal spatial scale [9] or using multi-temporal data [10]. Unmanned aerial vehicle (UAV), which has such benefits as strong maneuverability, high spatial resolution, low cost, and good real-time performance [11], provides a new remote sensing path for monitoring agriculture. In high resolution UAV remote sensing, PMF shows a clear spatial pattern (Figure 3, Section 2.2.1), which is different from that in satellite remote sensing. It suggests that the traditional handcrafted-features-based mapping methods may be not effective [12,13]. Specifically, PMF mapping for UAV remote images has very rarely been reported.

In 2015, the proposal of Fully Convolutional Network (FCN) [14] opened a new path for solving the dense label problem of deep learning for semantic segmentation [15], which greatly reduces the computational complexity and improves the accuracy of UAV remote sensing images [16,17]. For example, in reference [17], FCN outperformed traditional SVM and patch-based DCNN for classifying multi-view remote sensing images obtained by UAV. Moreover, other researches on FCN and UAV images were done, such as extracting information from infrared UAV data using Unet [18] and applying no-downsampling FCN to the problem of labeling high resolution images [19].

In this work, we built an end-to-end deep semantic segmentation method for accurately identifying the UAV-based PMF. We analyzed the experimental results of the spectral and spatial pattern between a classical SVM method [20] and two deep learning methods: FCN and its improvement, SegNet [21]. We found that the texture pattern of PMF for UAV remote sensing imaging is the determinant factor of plastic film monitoring. A combination of spectral and spatial features further improves the PMF identification. Generally, deep learning methods consistently outperform the traditional SVM method in all experiments in terms of precision and speed, which indicates that the proposed deep semantic segmentation model provides a very promising method of mapping PMF for UAV remote sensing images.

## 2. Study Area and Data Set

### 2.1. Study Area

The study was carried out in the Shahaoqu subdistrict of Hetao irrigation district, Inner Mongolia, China (Figure 1). This irrigation field is an independent unit in the northwestern Hetao with a total area of 52.4 square kilometers, located at 40°52′~41°00′N and 107°05′~107°10′E, and its shape is similar to a narrow inverted triangle. Its ground is relatively flat with elevation ranging from 1034 m to 1037 m. This region belongs to the middle temperate monsoon climate, mainly planted with sunflower, wheat, and corn, among which the sunflower-planting area is the largest, accounting for about 47.9%, whereas corn and wheat account for 23.6% and 12.6%, respectively. Because of low, unevenly distributed monthly precipitation, combined with huge temperature change [22], plastic film is widely used here.

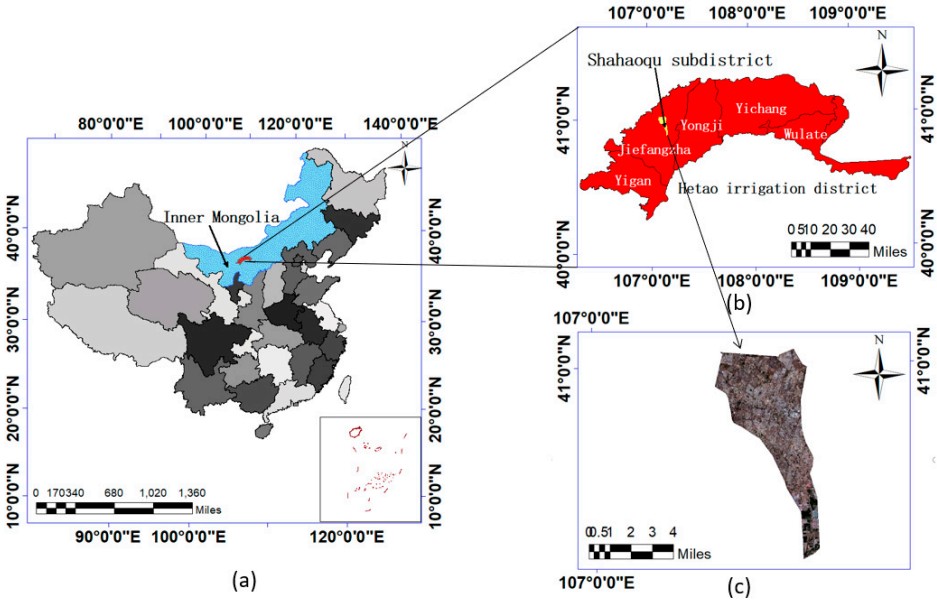

**Figure 1.** Location map of the study area. (**a**) Inner Mongolia, China; (**b**) Hetao irrigation district; (**c**) Shahaoqu subdistrict.

### 2.2. Data Set

#### 2.2.1. Data Sets Acquisition

Unmanned aerial vehicle remote sensing images were obtained at two different periods, 30 April–3 May and 13–17 August 2018. The acquisition devices were Jingwei M600 multi-rotor UAV (Figure 2a) produced by SZ DJI, China and the Micro-MCA multi-spectral camera (Figure 2b) produced by Tetracam, Inc., Chatsworth, CA, USA. The camera has 6 different channels including three visual bands whose central wavelengths are 490 nm (blue light), 550 nm (green light) and 680 nm (red light), and three near-infrared bands whose central wavelengths are 720 nm, 800 nm and 900 nm, where the width of each channel is 35 nm. Each band is equipped with a 1.3 million-pixel CMOS sensor and the resolution of each image is 1280 × 1024. After repeated tests, main flight parameters including altitude, time, and speed were set as 120 meters, 13: 00 and 9.2 m/s, respectively. To ensure the quality of images, the UAV hovered over the farmland for 5 to 10 seconds until the fuselage was stabilized and then the aerial vehicle photographed.

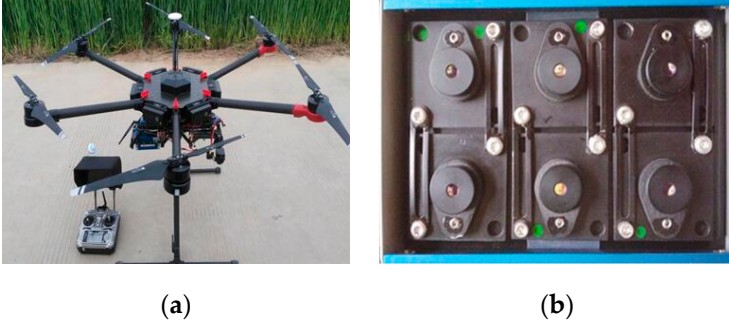

(**a**)　　　　　　　　　　　　　　　　　　　(**b**)

**Figure 2.** Unmanned aerial vehicle (UAV) and six-channel camera used in this study. (**a**) Jingwei M600 UAV; (**b**) Micro-MCA 6-channel camera.

In each experimental field, six original single-band images were extracted, registered, and synthesized by PixelWrench2 software, and the 6-band multispectral remote sensing images in TIF format could be derived. These aerial photos of UAV were corrected and spliced by pix4D. Then, ground control points were used to geometrically correct these photos to generate the final orthophoto image. After preprocessing UAV images, 18 non-intersecting 6-channel images with a size of 2560 × 2560 (pixel) were obtained altogether, whose spatial resolution is about 0.06 m.

### 2.2.2. Image Labeling and Analysis

After on-the-spot investigation, we used an open source software named Labelme [23] to manually annotate images. To analyze the factors that could interfere with the recognition accuracy, we first divided the PMF class into four subcategories: white film, black film, water-soaked film and soil between films, which are illustrated in Figure 3. At the same time, the non-PMF class (backgrounds) consists of five subcategories: road, green plants, crop-free farmland, waste land, and other components. The subcategory "other components" includes objects such as houses, water-contained irrigation canals, and rivers that are too small and complex to take into account. We selected 1000 pixels in each class randomly and drew the spectral curves (Figure 4) after taking the average value and confidence interval.

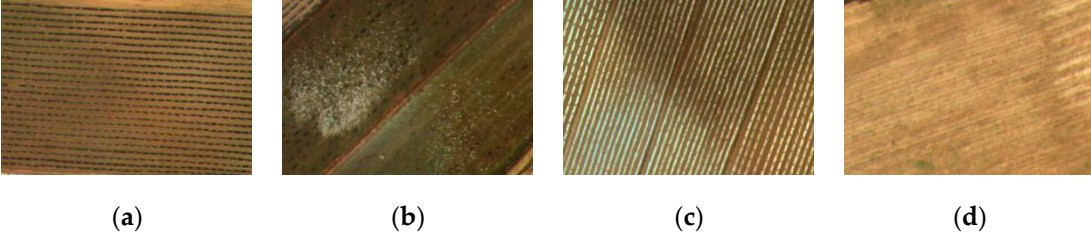

(**a**)　　　　　　　　　(**b**)　　　　　　　　　(**c**)　　　　　　　　　(**d**)

**Figure 3.** Typical kinds of Plastic Mulched Farmland (PMF) and background; (**a**) black PMF; (**b**) water-soaked PMF; (**c**) white PMF; (**d**) crop-free farmland. In addition, the soil between the plastic film stripes in (**a**–**c**) is also classified as PMF.

The spectral curves in Figure 4 convey a lot of information. First, the spectral curves of soil between film, water-soaked film, black film, and road are relatively close, which means that it is difficult to distinguish them correctly by spectral information alone, such as confidence intervals of water-soaked film and road overlapping each other in all six bands. Secondly, the spectral curve of white film is quite different from those of black film and water-soaked film, which indicates that there is a great difference in spectral information within the PMF class. In addition, it can be seen from remote sensing images that the texture of crop-free farmland illustrated in Figure 3d is similar to that of white film, because there are no crops growing in crop-free farmland at this time and they both have stripe-like texture.

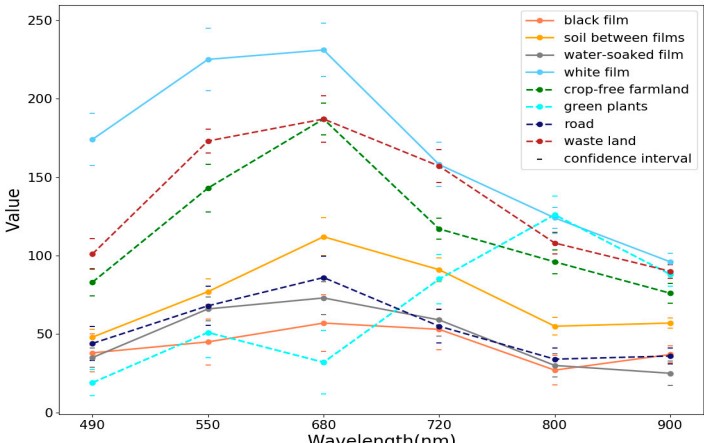

**Figure 4.** Spectral values and 95% confidence intervals of each subcategory on each band of UAV images (490 nm: Blue; 550 nm: Green; 680 nm: Red; 720 nm: Near-infrared; 800 nm: Near-infrared; 900 nm: Near-infrared).

## 3. Methods

The workflow of this study is presented in Figure 5. The preprocessing algorithms of UAV images, the feature extraction algorithm, and the deep learning algorithm were used in our study. In the beginning, UAV images were extracted, registered and orthorectified, as described in Section 2.2. These images were annotated and split into training images and test images. Then, training images and labels were used to train SVM (followed by the feature extraction stage), FCN, and SegNet, the step which outputs three models. Finally, these models were analyzed on test images by accuracy assessment method and then used to map PMF.

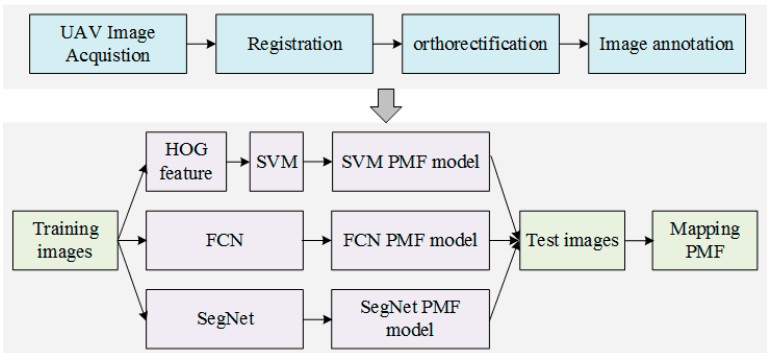

**Figure 5.** Workflow of the study. HOG: Histogram of Gradient; SVM: Support Vector Machine; FCN, SegNet: two deep semantic segmentation models.

### 3.1. Traditional SVM Method

For comparing with deep learning methods, we used a traditional SVM method to classify PMF. Histogram of Gradient (HOG) [24] was used to extract the object feature due to its wide application in computer vision.

### 3.1.1. Feature Extraction

Extracting appropriate features is crucial to traditional machine learning in image classification tasks because spectral, texture or spatial feature is the base for the following classification step. We cut 64 × 64 (pixel) small images from the 2560 × 2560 (pixel) images and extracted their HOG feature. All pixels in these small images belong to the same class. Then, each small image was labeled with 0 or 1 which indicates non-PMF or PMF, respectively. The 64 × 64 small image was sliced into 64 cells

whose size was 8 × 8. The gradient histogram of all 64 pixels was counted in each cell and it output a feature vector of 18 elements. Then, the adjacent 3 × 3 cells were merged into a block and, at the same time, feature vectors were linked. We scanned the 64 × 64 (pixel) image with a stride of 24 pixels, and all feature vectors of all blocks were linked to get the feature vector of 17,496 elements belonging to a 64 × 64 (pixel) image.

### 3.1.2. SVM Classification Method

Support vector machine is a machine learning method for binary classification and its basic model is a max marginal linear classifier. It has been widely used in the remote sensing field for classification tasks [25,26]. When the training data are linearly inseparable, the kernel function can be used to make it a nonlinear SVM [27], which is equivalent to implicit learning of linear SVM in a high dimensional feature space. The main idea of SVM is to solve the separated hyperplane which can correctly divide the data set and maximize the geometric interval between the positive and negative categories, which means classifying data. For the linearly separable data, there are infinitely linear hyperplanes but the separation hyperplane with the largest geometric interval is unique and has the best prediction ability to classify new examples.

### 3.2. Deep Semantic Segmentation Methods

### 3.2.1. Fully Convolutional Network

Convolutional Neural Network (CNN) rose in popularity because AlexNet [28] had achieved a higher score than traditional methods in ImageNet image recognition competition. It was soon introduced into remote sensing image classification tasks [12,29]. In the basic image classification tasks, CNN needs an image as the input, and after the convolution layer, activation function, pooling layer, and full connection layer, it outputs a value indicating its category. Due to the end-to-end procedure and the larger receptive field gotten by the convolution layer, CNN extracts the information of images better than the traditional machine learning method. Remote sensing classification differs from basic classification mainly in that each pixel in the image needs to be classified instead of the entire image, which is called the dense label problem. In the patch-based CNN [30,31], a small image of a certain size is cut out from the remote sensing image and put into CNN, and then the network will output the prediction of its central pixel. To achieve a mapping with the same size as the original remote sensing image, sliding window is applied on each small image cut out, and this process is repeated until every single pixel gets a prediction. Its inefficiency blocks up its utility in the dense label problem of semantic segmentation.

To solve this problem, Long et al. [14] proposed FCN in 2015. The main idea of FCN is one of making corresponding-sized dense predictions with an input of arbitrary size. As the first fully convolutional network for semantic segmentation, FCN has made three main changes to VGG-16: adapting the normal CNN, setting upsampling layer, and multi-scale feature fusions module (Figure 6). More details are provided below.

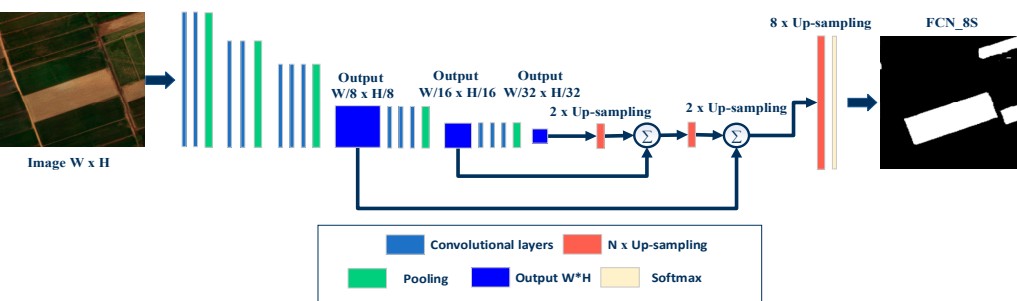

**Figure 6.** Network architecture of FCN_8s.

First, after re-architecting and testing AlexNet, VGG-16 and GoogLeNet, it was found that using VGG-16 as a backbone network can get better results from the PASCAL VOC 2011 data set, so the improved VGG-16 was used as the backbone network of FCN. To obtain the output corresponding to the input image size, the traditional fully connected layer at the bottom of VGG-16 was replaced by a deconvolution layer. Because FCN does not contain a fully connected layer, its trained model can adapt to the input of any size.

Secondly, because of five max-pooling layers of VGG-16, after the modification mentioned above, the output of the network was changed to only 1/32 of the input size, whereas the semantic segmentation task requires that the output size corresponds to the input size to make a prediction for every pixel. Therefore, feature maps at the bottom of FCN need to be upsampled to obtain the pixelwise prediction. The upsampling method adopted by FCN is a trainable deconvolution. In a common convolution operation, the output of $i \times i$ image processed by a $k \times k$ convolution kernel is $(i - k + 1) \times (i - k + 1)$ without padding or $i \times i$ with padding. In a deconvolution process that is the inverse process of convolution, the output of $i \times i$ image processed by a $k \times k$ deconvolution kernel with s stride (insert s − 1 zeros into each 2 computational elements) is $s \times (i - 1) + k$ square. The deconvolution layer is trainable and enables nonlinear learning with the help of the activation function. Thus, they are more effective than simple bilinear interpolation which is a popular upsampling method.

Third, it is clear that the direct use of deconvolution to process the output information of the upper layer can only get rough results; in order to solve the problem of rough edges and loss of detailed information after upsampling, FCN adopts a jump structure that can fuse multi-scale features to combine deep-level information with low-level information. Low-level features are full of location information, which can tell the network where it is, whereas deep-level features are full of semantic information, which can tell the network what class each pixel belongs to. To obtain more accurate and detailed segmentation results and to ensure the robustness of the model, FCN concatenates low-level feature maps and high-level feature maps as input of the deconvolution layer before the final results are outputted.

The FCN_8s model, which reduces the size of input images to 1/8 before the last upsampling step, has the best effect among the three proposed structures (FCN_8s, FCN_16s, FCN_32s) in reference [14]. Because FCN_8s combines more lower-level features than the other two models, its processing effect on the edge of objects is better than those of the FCN_16s model and FCN_32s model, so we only used the FCN_8s model in our paper.

### 3.2.2. SegNet

SegNet [21] is a fully convolutional network based on encoder-decoder architecture proposed by Badrinarayanan. It is composed of an encoder network, a decoder network and a pixelwise classifier. Aiming at solving the rough prediction problem produced by the previous semantic segmentation methods, this network aims to extract effective features to obtain an accurate boundary location. In view of the defects of some existing fully convolutional networks (such as FCN and DeconvNet [32]), SegNet has made some improvements.

First, the encoder network of SegNet reserves pooling indices, locations where retained values of pooling windows in the pooling layer exist. A max-pooling layer is widely used in neural network processing images because of its function in retaining translation invariance and reducing trainable parameters. However, when it is decreasing the number of data transmitting to the next layer, it is also removing a lot of boundary information (for a max-pooling layer with a parameter value of n, the number of data transmitting to the next layer is only $1/n^2$ of the layer's input), which leads to a coarse prediction. In addition, the method reserving the whole feature map before pooling layers, which is adopted by FCN, consumes too much memory space and greatly increases the amount of computation, so it has a low productivity in relation to practical problems. Therefore, SegNet is designed to preserve the benefits of max-pooling while maximizing the retention of boundary information with minimal memory space and computing time. To achieve this goal, SegNet reserves

the locations of values which are the outputs of max-pooling layers within the encoder network, i.e., the locations of all the maximum values in every max-pooling window. Through reserving these pooling indices, the boundary information of different abstraction levels is preserved and the cost of this process is reduced at the same time.

Second, the trainable deconvolution method results in a large number of trainable parameters, and the decoder network of SegNet needs to use the locations transmitted from the corresponding layer in the encoder network to obtain boundary information (Figure 7). Therefore, in the upsampling layer in the decoder network, SegNet places each cell of the inputted w × w feature map into the outputted 2w × 2w feature map according to pooling indices, which is also called unpooling. Because in each 2 × 2 unpooling window (we used 2 × 2 max-pooling, so the unpooling window's size is also 2 × 2) only one cell is filled with a corresponding value, whereas the other 3 cells are filled with zeros, the outputted feature map is sparse. Thus, in order to generate a dense feature map, several channel-reducing convolution layers are added behind each unpooling layer.

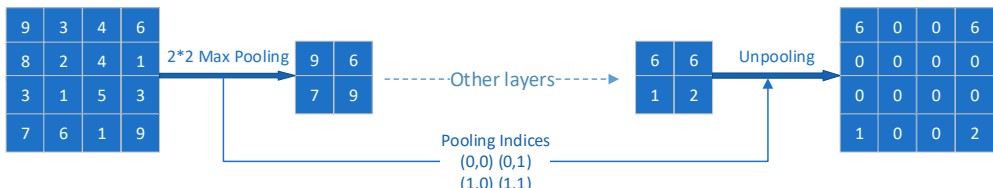

**Figure 7.** Max pooling and unpooling processes in SegNet.

The most important part of SegNet is the design of its decoder network, which is different from other networks (both FCN and DeconvNet use deconvolution to decode). It can explain the image boundary information well while reducing the memory space and computing time. The architecture of SegNet is shown in Figure 8.

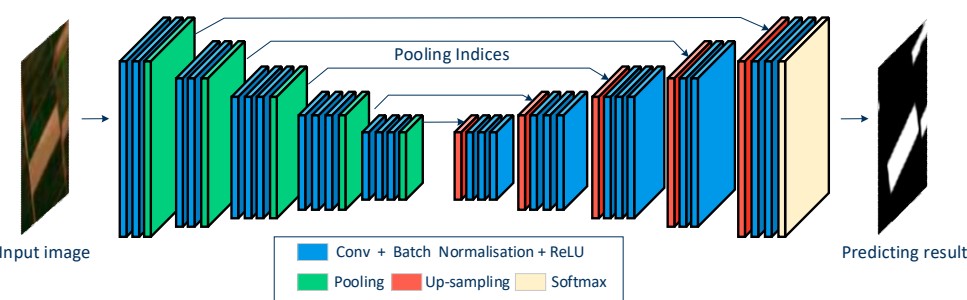

**Figure 8.** Network architecture of SegNet.

### 3.2.3. Network Training

Considering the large amount of data and long computing time of UAV images using deep learning methods, the NVIDIA GEFORCE GTX 1070 (8 GB memory) is used to train models. Cross entropy is chosen as the loss function to measure the predicted mapping's deviation from the mapping of our manual annotation. Please refer to reference [33] for more details In basic tasks, each image produces a final value corresponding to the category to which the image belongs, but the number of tags generated by the fully convolution network is equal to the number of pixels in the input image, which is a dense label. Therefore, in our network, the total loss is the mean of the cross-entropy loss of each pixel in the result mapping. The formula is defined as

$$J = -\frac{1}{hw}\sum_{i=1}^{h}\sum_{j=1}^{w}\left[y_{ij}\ln\alpha_{ij} + (1 - y_{ij})\ln(1 - \alpha_{ij})\right] \tag{1}$$

where $i$ and $j$ are the coordinates of pixels, $y_{ij}$ and $\alpha_{ij}$ indicate the true value and prediction value of the pixel located at $(i, j)$, $h$ and $w$ indicate the height and the width of image, respectively. In this study, $y_{ij}$ and $\alpha_{ij}$ are both binary parameters of either 0 or 1, indicating which class the corresponding pixel with row number $i$ and column number $j$ belongs to. For our methods, stochastic gradient descent (SGD) was selected to update weights, and the learning rate was set to 0.01.

In our experiments, 12 out of 18 images were selected to make a training set and verification set. To make full use of the information in the images and taking factors such as GPU memory size into account, we randomly selected (x, y) coordinates in 12 training images and cut out 100,008 images with 256 × 256 (pixel) size. After 25% rotation, adding noise and mirroring, the training set contained 75,006 images and the verification set contained 25,002 images. Each batch included 8 pictures during the training. The network trained 15 epochs on the training dataset to get 15 models, within which the model with the highest score on the test set was taken as the final model of the network.

### 3.2.4. Model Test and Accuracy Assessment

The remaining six images were used as the testing data set. Because the input and output of the model were both images of 256 × 256 pixels, the sliding window method was adopted. We cut 256 × 256 (pixel) images from top to bottom and from left to right and put them into the trained model. There was a 50% overlap rate between the two small images adjacent, and the voting mechanism was applied to the result of the overlapping part.

Generally, pixel accuracy (PA, which is calculated by dividing the number of truly predicted pixels by the number of all pixels) is the evaluation index in the classification problem, but when applied to the semantic segmentation problem, it showed a low sensibility to the errors of a few pixels, such as the boundary. Therefore, mean intersection over union (mIoU) was chosen to be the index evaluating our models, which is sensitive to a few pixel errors in result mapping and is also a common indicator of semantic segmentation. The formula is defined as

$$mIoU = \frac{1}{m}\sum_{i=1}^{m}|A_i \cap B_i|/|A_i \cup B_i| \tag{2}$$

where $A_i$ and $B_i$ indicate the area of $i$ class in label and the area of $i$ class in result mapping, respectively, and m indicates the number of classes.

## 4. Results

### 4.1. PMF Identification Only with Texture Feature

As mentioned above, the high resolution of UAV remote sensing images makes a spatial pattern of PMF, clearly distinct from other components. Therefore, as described in this section, we built the model by using a single-band image only with texture feature to classify the PMF. Table 1 shows the PMF classification results of SVM, FCN, and SegNet for six fields, where the first two best results for each classification method are listed. Figure 9 presents the segmentation results of four fields.

**Table 1.** Results of PMF identification for single-band UAV remote sensing image.

| Methods | Best 2 Bands (nm) | Accuracy (mIoU, %) | | | | | | | Time (/Field) |
|---------|-------------------|--------|--------|--------|--------|--------|--------|---------|---------------|
| | | Field1 | Field2 | Field3 | Field4 | Field5 | Field6 | Average | |
| SVM | 720 | 75.35 | 60.43 | 57.44 | 78.38 | 93.76 | 70.35 | 72.62 | 2 h 17 m |
| | 800 | 59.50 | 66.31 | 65.15 | 76.88 | 95.42 | 69.66 | 72.15 | 2 h 15 m |
| FCN_8s | 800 | 93.51 | 81.07 | 78.80 | 86.89 | 96.84 | 86.46 | 87.30 | 10.09 s |
| | 900 | 85.31 | 83.32 | 82.94 | 88.51 | 92.12 | 87.21 | 86.57 | 10.13 s |
| SegNet | 800 | 96.38 | 83.38 | 82.00 | 93.39 | 93.66 | 83.15 | 88.66 | 16.50 s |
| | 900 | 95.39 | 83.26 | 84.12 | 93.05 | 91.37 | 84.90 | 88.68 | 16.44 s |

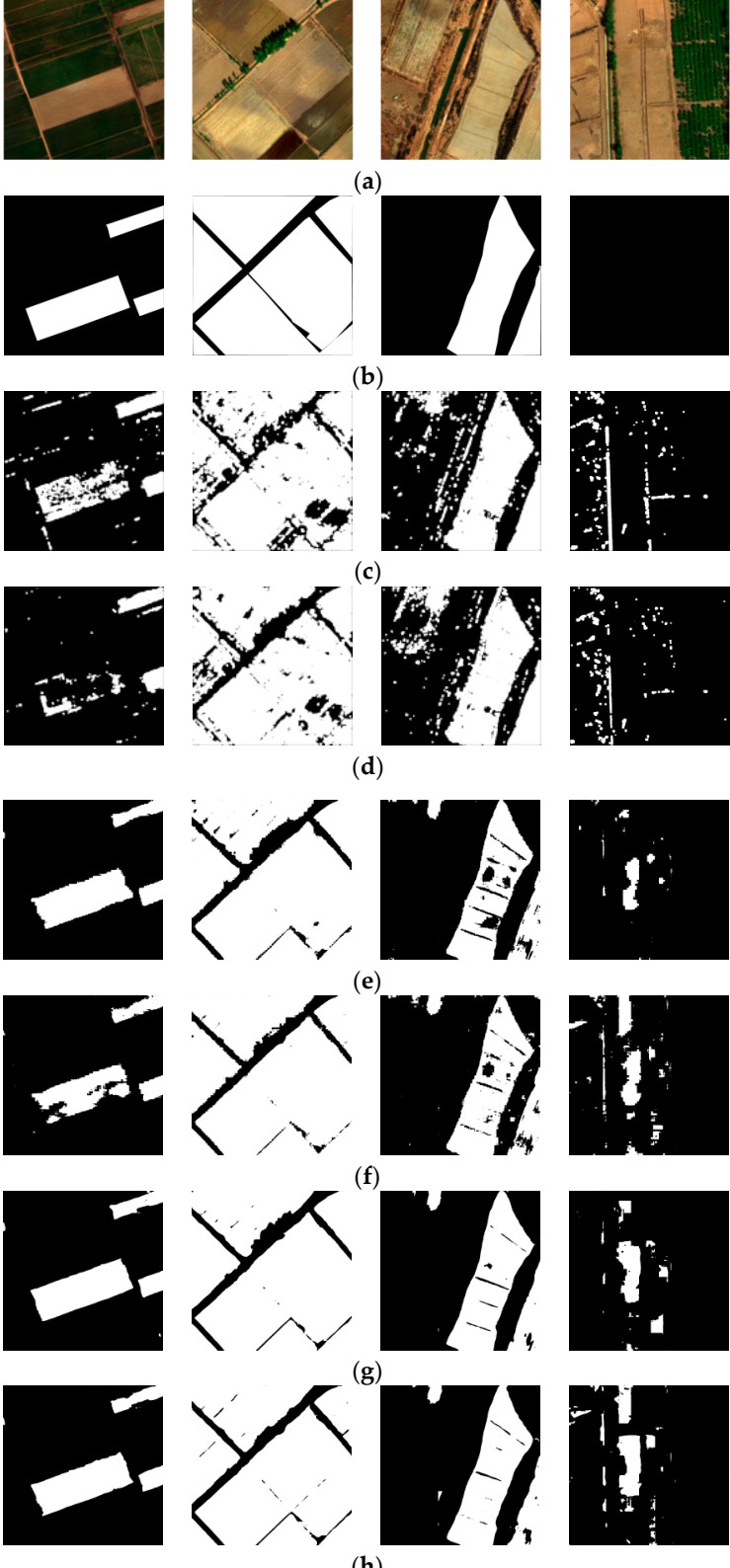

**Figure 9.** Plastic mulched farmland identification results for single-band UAV remote sensing image of field 1, field 3, field 4, and field 5. (**a**) Test images consisting of three fields; (**b**) ground truths corresponding to test images; (**c**) SVM with 720 nm band; (**d**) SVM with 800 nm band; (**e**) FCN_8s with 800 nm band; (**f**) FCN_8s with 900 nm band; (**g**) SegNet with 800 nm band; (**h**) SegNet with 900 nm band.

From Table 1, we can see that for the SegNet model, the near-infrared band whose central wavelength is 900 nm outperforms other bands at an accuracy of 88.68%, followed by the near-infrared band, whose central wavelength is 800 nm at an accuracy of 88.66%. In the case of using the FCN_8s model, the best two results are achieved by two near-infrared bands whose central wavelengths are 800 nm and 900 nm at an accuracy of 87.30% and 86.57%, respectively. When applying single-band data to the SVM method, the band whose central wavelength is 720 nm outperforms other bands at an accuracy of 72.62%, followed by the 800 nm band whose accuracy is 72.15%. Besides, FCN_8s is the fastest method, taking only about 10 s to map a field, followed by SegNet, whose time consumption is about 16 s per field.

It can be seen in Figure 9 that identifying results only with single-band data shows a coarse boundary. However, the boundary of FCN (Figure 9e,f) and SegNet (Figure 9g,h) is smoother than that of SVM (Figure 9c,d) which indicates an advantage of deep learning methods over the traditional method. For SegNet, thanks to its ability to extract edge information efficiently, its segmentation boundary is the smoothest.

## 4.2. PMF Identification Using Multiple-Band Images

Although the main factor that separates PMF from other components is its characteristic spatial pattern, the combination of texture feature and spectral feature can increase the classification accuracy. Table 2 lists the results of PMF classification by using RGB three bands and all six bands for three methods. Figure 10 illustrates the segmentation results for four fields of three models using 3-band and 6-band data.

**Table 2.** Results of PMF identification for multiple-band UAV remote sensing image.

| Methods | Number of Bands | Accuracy (mIoU, %) | | | | | | | Time (/Field) |
|---------|-----------------|--------|--------|--------|--------|--------|--------|---------|---------------|
| | | Field1 | Field2 | Field3 | Field4 | Field5 | Field6 | Average | |
| SVM | 3 | 90.45 | 66.27 | 64.43 | 72.65 | 96.6 | 72.03 | 77.07 | 5 h 3 m |
| | 6 | 92.34 | 67.66 | 76.14 | 81.45 | 96.3 | 74.27 | 81.36 | 8 h 31 m |
| FCN_8s | 3 | 92.42 | 76.44 | 70.69 | 78.16 | 97.77 | 70.97 | 81.08 | 10.83 s |
| | 6 | 96.51 | 85.29 | 81.97 | 91.2 | 99.65 | 83.54 | 89.69 | 11.15 s |
| SegNet | 3 | 96.99 | 84.51 | 80.77 | 93.37 | 99.94 | 81.85 | 89.62 | 17.37 s |
| | 6 | 97.35 | 85.53 | 82.41 | 96.65 | 99.70 | 81.94 | 90.60 | 17.92 s |

Table 2 shows that when using 3-band data, the accuracy of SegNet is 12.55% higher than that of SVM and 8.54% higher than that of FCN_8s, which is 4.01% higher than that of SVM. When 6-band data is used, the accuracy of SegNet is 9.24% higher than that of SVM and 0.91% higher than that of FCN_8s, which is 8.33% higher than that of SVM. From the above results, we can conclude that the best results are obtained by using a SegNet model with 6-band data. Combined with the results in the spatial pattern analysis section, we can see that compared to single-band data, 3-band data and 6-band data generally bring higher accuracy to all three methods, especially on Field5 using SegNet, whose average accuracy of 3-band data is 8.57% higher than that of single-band data, and that of 6-band data is 8.33% higher than that of single-band data. This is mainly because, in Field5, with 3-band or 6-band data, SegNet mistakes less crop-free farmland for PMF. For 3-band data and 6-band data, FCN shows the fastest prediction speed, with a time consumption of about 11 s per field, followed by the 17 s per field of SegNet.

From Figure 10, we can see that the mapping boundary of SegNet (Figure 10e,f) is the smoothest among 3 methods with both 3-band data and 6-band data. Adding three near-infrared bands greatly improves the mapping boundary of SVM_HOG (Figure 10a,b) and FCN_8s (Figure 10c,d). However, the improvement to SegNet is not as obvious as that of the other two methods.

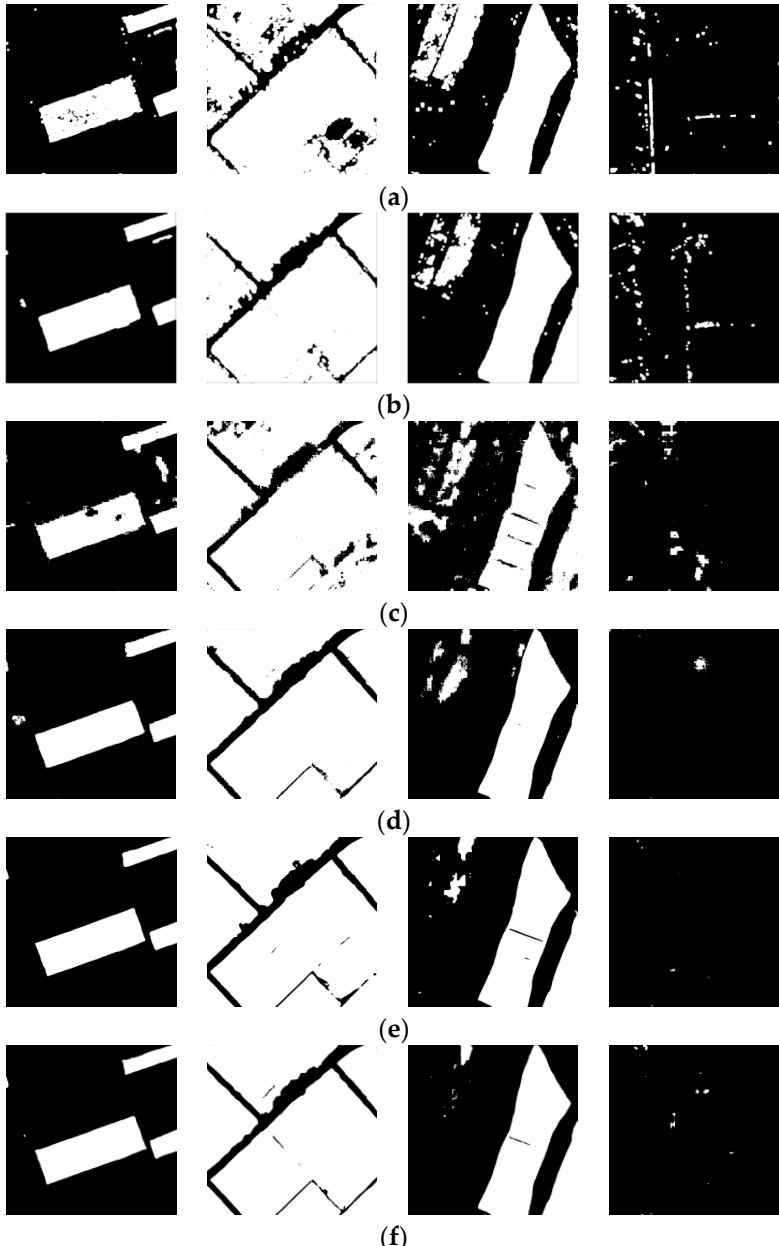

**Figure 10.** Segmentation results of PMF identification for multiple bands UAV remote sensing image of field 1, field 3, field 4, and field 5. (**a**) SVM with 3-band data; (**b**) SVM with 6-band data; (**c**) FCN_8s with 3-band data; (**d**) FCN_8s with 6-band data; (**e**) SegNet with 3-band data; (**f**) SegNet with 6-band data. In addition, test images and ground truth are shown in Figure 9.

## 5. Discussion

Unmanned aerial vehicle PMF mapping technology has great application potential in environmental improvement and precision agriculture, but there are also some challenges in its application. In this study, different features extracted by different methods with single-band, 3-band, and 6-band UAV data were used to map PMF images. With regard to the experimental results above, we discuss the following:

### 5.1. Contribution of Texture Feature

The high spatial resolution in UAV images and the inherently regular change of appearance of PMF result in its characteristic spatial pattern, i.e., the texture feature of PMF illustrated in Figure 3.

However, how the evident texture feature affects the PMF mapping task has not been examined. To explain its effects, a comparative analysis was implemented using texture feature extracted from 6 bands independently, whose central wavelengths were 490 nm, 550 nm, 680 nm, 720 nm, 800 nm and 900 nm.

By building models only with single-band data for three methods, we received satisfactory PMF segmentation results whose lowest accuracy was more than 70% and the highest was over 88%. Because single-band data has hardly any spectral information, it is obvious that texture information plays a very important role in the PMF recognition task for UAV images. Besides, the bands for which we got the best 2 results of three methods were all near-infrared bands, so near-infrared bands were more suitable for this segmentation task than RGB bands when only single-band data was used. However, by applying single-band data to our methods, the edges of these segmentation results were coarse on the whole, which means that the texture feature extracted from only one band could not provide enough information for a smooth-edge segmentation.

*5.2. Combination of Spectral and Texture Feature*

After discovering that the texture feature extracted from single-band data produces a satisfactory accuracy but a coarse edge, two comparative experiments combining spectral and texture feature extracted from three visual bands and 6 bands, respectively, were used to examine the effects of adding spectral information. Like the single-band experiment, we built two deep learning models and a traditional SVM model to get a credible comparative result.

By adding more spectral information, the three methods performed better, especially SVM. For FCN, when 6-band data was used, the accuracy increased greatly, mainly because it mistook less crop-free farmland and wasteland for PMF, such as in field 4 (Figure 10) and field 6. However, for the SegNet model, from its results with single-band data and 3-band data, we can see that this model was able to extract texture information very well. Thus, the augmentation of spectral information brought about by adding 3 near-infrared bands only slightly improved its performance. Besides, by using more spectral feature, the segmentation edge became smoother from single-band data to 3-band data and from 3-band data to 6-band data.

*5.3. Advantage of Deep Learning over Traditional SVM*

From previous works on deep learning methods applied in remote sensing, it is concluded that deep learning methods significantly outperform traditional machine learning methods [17,29]. However, the extent of the outperforming is not known when it comes to PMF mapping for UAV images. To highlight the advantages of the methods proposed in this study, a comparison of accuracy, edge smoothness, and mapping speed between two deep learning methods and the traditional SVM method was conducted using single-band data, 3-band data and 6-band data.

By comparing the performance of three methods, we found that SegNet outperformed the other two methods both in accuracy and smoothness of segmentation. Compared to block-wise-based classification in the traditional SVM method, deep semantic segmentation based on pixel-wise classification was much faster than SVM. In application, the selection of features extracted from images was the basis of traditional methods, and this step, which should be done according to researchers' experience and professional knowledge, always costs massive labor. However, deep learning methods that extract optimal features automatically can save a lot of time and reduce the requirement for expertise.

SegNet is slightly slower than FCN because its decoder architecture brings more computational burden. In addition, in order to adapt the proposed models to three-band or six-band UAV remote sensing images, only the third dimension of kernels in the first convolution layer in our models was changed according to the channels of input data. Therefore, the increase of channels of input data results in only slightly more computational burden for predicting time.

*5.4. Differences from Existing Works*

For satellite remote sensing images with low and middle resolution, such as Landsat TM, Landsat 8, and MODIS-NDVI, PMF pixels mix the information from plastic mulched farmland and soil between PM stripes and form their characteristic spectral features distinguished from other classes. Several previous works [5,7,8,10] showed that spectral features played a dominant factor in mapping PMF, whereas combination of spectral and texture feature can raise the classification accuracy. However, due to more detailed spatial structural information obtained by high-resolution (2 m × 2 m) remote sensing from GaoFen-1 satellite, recent work [9] showed that the texture feature plays a more important role in PMF identification than that in middle and low spatial resolution data via appropriate spatial scale experiments.

Moreover, in our UAV remote sensing image with a higher spatial resolution of 0.06 m × 0.06 m, the soil between PMF stripes is clearly visible, and has a finer spatial structure than those from GaoFen-1 used in reference [9]. Obviously, the spectral feature alone is difficult to correctly classify the soil between PPF stripes, and texture feature can handle it well, as illustrated in our work.

For the classification method, in previous works [5,7–10], spectral and texture features manually extracted by traditional methods such as Normalized Difference Vegetation Index (NDVI), Gray-Level Co-occurrence Matrix (GLCM), and HOG have lower representation capability than deep learning feature. By fully exploiting effective end-to-end feature learning capability, our deep semantic segmentation methods consistently outperformed traditional SVM with handcrafted feature in terms of accuracy and speed.

*5.5. Limitations and Future Work*

Deep semantic segmentation methods need a great amount of computational resource, including high-end GPU and more training data compared to traditional methods, to learn a robust classification model. Recently, some light CNN models [34] with fewer parameters and good performance have received much attention, which may alleviate this problem to some extent.

In terms of data acquisition, the PMF in the UAV images is of a single growth period and from only one region. Besides, only the limited agricultural area is sampled and some other types of land use, such as forest and city, may affect the results of this study. In the future, we will extend the proposed methods of UAV remote sensing images to more regions with different growth periods to obtain more precise plastic mulched farmland mapping.

## 6. Conclusions

Although the application of plastic mulch has improved the quality and quantity of agricultural production, it has brought great pressure to our environment. Thus, fast and accurate PMF identification is important for monitoring its spatial and temporal distribution and providing a reference for agricultural production. In this paper, we built the deep semantic segmentation models to identify PMF through high-resolution UAV remote sensing images. We used fully convolutional networks, including FCN and SegNet, to automatically extract deep semantic features and realize end-to-end learning and prediction with higher accuracy and faster speed than the traditional SVM method. From our experiments, we draw the following conclusions:

First, the PMF's spatial pattern is a key factor in separating it from the background, which is verified by three classification methods, including traditional SVM and two deep semantic segmentation methods, through the experiments on only single-band UAV remote sensing images.

Second, we used multiple-band data with spectral and spatial feature providing more information on PMF to improve the classification accuracy of PMF. We found that all algorithms tested on six-band data performs better than a single band or three bands.

Finally, compared to traditional SVM classification, the proposed deep semantic segmentation methods in this work achieved an end-to-end PMF identification with higher accuracy and faster

speed due to their advantages of strong feature representation and direct pixel-based classification. The proposed SegNet model, whose network architecture in the max-pooling and unpooling stage was superior to FCN, leads to better predicting accuracy in all experiments among the three classification methods.

**Author Contributions:** Q.Y. and M.L. designed and conducted the experiments and performed the programming works. Q.Y. and J.N. contributed extensively to the manuscript writing and revision. J.N. and Z.Z. provided suggestions for the experiment design and data analysis. S.Y. and M.L. contributed to the editing of the manuscripts. W.H. and J.N. supervised the study.

**Acknowledgments:** This work was supported by the National Natural Science Foundation of China under grants 61876153 and 31501228, the National Key Research and Development Program of China (2017YFC0403203 and 2016YFD0200700), and the Fundamental Research Funds for the Central Universities under grant 2452019180. We thank Xiaoqing Fan and Hui Liang for labeling data.

**Conflicts of Interest:** The authors declare no conflict of interest.

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
