# Peer review of "Mapping Plastic Mulched Farmland for High Resolution Images of Unmanned Aerial Vehicle Using Deep Semantic Segmentation"

_remotesensing, doi:10.3390/rs11172008_

Round 1

Reviewer 1 Report

Authors presented the idea that has great potential and can be useful in agriculture. Although its great potential, there are some (future) challenges. 

The first consideration is low accuracy, is ~90% accuracy enough for its intended application? 

Authors stated (line 405) “Previous work concluded that spectral features play a more important role in PMF mapping than texture features for satellite remote sensing images [5]. However, the high spatial resolution in UAV images results in its characteristic spatial pattern… How the evident texture feature affects the PMF mapping task hasn’t been examined. To  explain its effects, a comparative analysis is implemented using texture feature extracted from 6 bands”

My second consideration is sensor involved. By using a drone instead of satellite images, they obtained higher resolution images, and texture feature could be involved. Additionally, in several statements, authors comment that results from 6 channels are better than from 3 channel (which is no surprise at all). Another author's comments are that bets results are obtained with near-IR bands (~900 nm). 

My question is, are authors favoring texture features instead of spectral features? 

Wold usage of SWIR (alone or as the additional channel) be more beneficial for the proposed method? What about the hyperspectral camera (usually used in agriculture monitoring)?

As can be seen from figure 3, differences from similar materials (water-soaked plastic, black plastic) are minor. Please add confidence intervals to the presented graph. Is there an interval where materials could be completely mismatched?

Also please add information about camera waveband for each channel, to wide bands may decrease its performance. 

Line 163, HOF is not defined.

In conclusion, the authors presented interesting work, but the main consideration arises from the sensors involved, which finally results with possibly unacceptable accuracy. 

Reviewer 2 Report

General: In this study, an end-to-end deep learning method is proposed, to recognize plastic mulched farmland for Unmanned Aerial Vehicle remote sensing images, by comparing two deep semantic segmentation methods, SegNet and FCN, and traditional classification method-Support Vector Machine.

I think this is an interesting work dealing with the recognition of plastic mulched farmland, the widespread use of which results in a series of environmental problems. The introduction is relevant and theory based. The methods are generally appropriate although poor English language makes the reading often difficult. In general, English language is not of publication quality and requires major improvement. The results are clearly presented. In Discussion, the authors could accompany their findings with appropriate references. Detailed comments and suggestions are reported below:

1.    Page 1, abstract: Include a conclusion statement at the end of the abstract.

2.    Page 3, line 81: "Moreover, Some other..." replace capital s.

3.    Page 4, section 2.1: The authors describe the study area in detail. However, adding a location map would be valuable for identifying the study area.

4.    Section 2.2 and 3: Check present and past tenses in the Data and Methods sections. E.g. page 4, lines 123-125: “In each experimental field, six original single-band images were extracted, registered and synthesized by PixelWrench2 software, and the 6-band multispectral remote sensing images in TIF format could be derived. These aerial photos of UAV are corrected and spliced by pix4D.” It is preferable “were corrected”.

5.    Page 4 to page 16: English language (syntax, grammar, tenses) needs extensive editing in sections 2, 3, 4 and 5.

6.    Page 4, line 128: “pixel size of 2560×2560” Please define the pixel size units.

7.    Page 5, line 138: “…film and soil between films, which are illustrated in Figure 3” PMF subcategories are shown in Figure 2 and not in Figure 3.

8.    Page 5, lines 140-141: “The subcategory other components includes objects such as houses and water source that are too little and complex to take into account.” What do you mean by “water source”? Rephrase it or be more specific. Moreover, check English language.

9.    Page 5, line 142: “…randomly and draw the spectral curves (Figure 2)” spectral curves are illustrated in Figure 3 and not in Figure 2.

10. Page 5, line 147: Spectral curves are illustrated in Figure 3 and not in Figure 2.

11. Page 5, line 152: “the texture of crop-free farmland illustrated in Figure 3d” correct the figure to 2d.

12. Page 6, section 3. Methods: For further clarification of the methodology, the authors could create a flowchart ordering the methods and data used in a graphical way.

13. Page 6, line 163: “…64 × 64 small images…” 64x64 pixel size or area? Please define it as well as the units. Correct it in any other image size is mentioned in the section 3.1.1. and lines 293, 299.

14. Page 9, lines 272-274: The authors refer to results which have not still presented. It should be more appropriate to discuss it in section 5.

15. Page 11, lines 330-333: The corresponding figures could be added while comparing the results of the different methods e.g. “However, the boundary of FCN (Figure 7e and 7f) and SegNet (Figure 7g and 7h) is smoother than that of SVM (Figure 7c and 7d) which indicates an advantage of deep learning methods over the traditional method.”

16. Page 13, lines 376-379: The corresponding figures could be added while comparing the results of the different methods e.g. “…mapping boundary of SegNet is the smoothest (Figure 7e and 7f)...”

17. Page 15, section 5.1: I think a more in-depth discussion of the contribution of texture feature, and accompany this with some references to other studies that demonstrate this contribution, would be helpful.

18. Page 16, section 6: In Conclusions, the authors could discuss the importance of recognizing plastic multched farmland, the contribution to the solution of environmental problems, and as a result the importance of their work.

Reviewer 3 Report

In this paper, the authors claim that through comparison among two deep semantic 25 segmentation methods including SegNet and FCN, and traditional classification method-Support 26 Vector Machine (SVM), they propose an end-to-end deep learning method aiming at accurately 27 recognizing PMF for UVA remote sensing images from Hetao irrigation area, Inner Mongolia, 28 China. 

Although the justification of the research is well defined, I could not find how the authors defined the formula for the total loss which is the sum of the cross entropy loss of each pixel in the result mapping. Can the authors analyse how their formula is based on :

h, w, α, are not defined.

Can the authors comment on the disadvantages of the semantic segmentation of aerial images using deep learning? (i.e. high computational power, High end GPU)?

Round 2

Reviewer 2 Report

Dear Authors,

Well done in revising the manuscript.

I suggest only one more correction, to delete the message appearing in Figure 1a that is not necessary.

Wish you all the best.

This manuscript is a resubmission of an earlier submission. The following is a list of the peer review reports and author responses from that submission.